# Dendrimers, an Emerging Opportunity in Personalized Medicine?

**DOI:** 10.3390/jpm12081334

**Published:** 2022-08-19

**Authors:** Anne-Marie Caminade

**Affiliations:** 1Laboratoire de Chimie de Coordination (LCC), CNRS UPR8241, 205 Route de Narbonne, CEDEX 4, 31077 Toulouse, France; anne-marie.caminade@lcc-toulouse.fr; 2LCC-CNRS, Université de Toulouse, CNRS, 31077 Toulouse, France

**Keywords:** dendrimers, clinical trials, phases 1, 2, and 3, polylysine, polyamidoamine

## Abstract

Dendrimers are highly branched macromolecules tailorable at will to fulfil precise requirements. They have generated a great many expectations and a huge number of publications and patents in relation to medicine, including in relation to personalized medicine, but have resulted in very poor clinical translation up to now. As clinical trials are the first steps in view of developing new compounds for (a personalized) medicine, this review focusses on the clinical trials carried out with dendrimers. Many of these clinical trials have been recently posted (2020–2022); thus, only very few concern phase 3. The safety and efficiency of essentially two main types of dendrimers, based on polylysine and polyamidoamide scaffolds, have been assessed up to now. These dendrimers were tested with the aim of treating mainly bacterial vaginosis, cancers, and COVID-19.

## 1. Introduction

Dendrimers [1] are a special class of synthetic macromolecules, constituted of branches built, step by step, around a central multifunctional core. Each layer of branching points creates a new “generation”. Most of the properties of dendrimers depend on the type of their terminal functions (Figure 1). Dendrimers are often considered as 3-dimensional soft nanoparticles [2], in opposition to hard metal nanoparticles. Despite the fact that nature has favored branching structures at all levels, from galaxies to trees and to dendritic cells, examples of branching at the molecular level are extremely rare. One can cite glycogen, a branched polymer of glucose [3], and also some cases of branched lignin [4], but none of them have a precisely highly branched structure, as do dendrimers. Such unusual structure has generated a many expectations for dendrimers: a huge number of publications and patents exist in relation to medicine, including in relation to personalized medicine (some recent reviews about dendrimers in relation to personalized medicine: [5,6,7,8,9,10,11,12]) but have resulted in very poor clinical translation up to now [13]. 

It must be said that these synthetic macromolecules, which do not resemble anything natural or biological, arouse a certain number of fears. The very first dendrimers were synthesized in 1978 [14], and their synthesis was mainly developed in the 80th [15,16,17] and the 90th [18,19,20,21]. However, since that time, only very few clinical trials have concerned dendrimers, as illustrated by the fact that the site “ClinicalTrials.gov” (accessed on 2 August 2022) [22] contains 423,241 research studies (2 August 2022), of which only about 26 concern dendrimers, many of them having been posted in 2020–2022. As clinical trials are the first steps in view of developing new compounds for (a personalized) medicine, this review will focus on the clinical trials carried out with dendrimers. Information is mainly taken from ClinicalTrials.gov (accessed on 2 August 2022) (USA) and in part from clinicaltrialsregister.eu (European Union) [23]. It should be noted that the word “dendrimer” is not always used in these clinical trials. A code to define the structure is used in most cases, which renders the search relatively difficult.

## 2. Clinical Trials with Dendrimers Based on Poly-L-Lysine

The very first large dendrimers synthesized and patented were based on poly-L-lysine [15], and the first example of clinical trials with dendrimers concerned poly-L-lysine dendrimers. They were first functionalized with 24 macrocyclic complexes of gadolinium on the surface (Figure 2A), affording a high-relaxivity agent for MRI (magnetic resonance imaging) [24]. This compound, named Gadomer-17 or SH L 643 A (Schering AG, Berlin, Germany), was first injected to healthy volunteers for imaging their aortoiliac region and then to patients with coronary artery disease, affording an improved detection [25]. However, there is no recent result published with this compound.

A second example of poly-L-lysine dendrimer was proposed by Starpharma (Australia) as an active ingredient in gels against sexually transmitted diseases, as it has antiviral properties and is able to block bacteria when formulated as a mucoadhesive gel [26]. It is based on a generation-four dendrimer bearing 32 Sodium 1-(Carboxymethoxy)naphthalene-3,6-disulfonate on the surface, named SPL-7013 as well as Astodrimer sodium (Figure 2B). The formulated gel, named VivaGel^®^, has been tested in at least 13 clinical trials for assessing first the safety, tolerability, PK, and acceptability of vaginal administration. The first conclusion of a randomized, double-blind, placebo-controlled, dose-escalation trial conducted in 2004 at the Royal Adelaide Hospital (Australia) was that VivaGel® applied vaginally once daily for 7 days at concentrations of 0.5% to 3% was safe and well-tolerated in women, with no evidence of systemic toxicity or absorption [27]. Another phase 1 clinical trial (registered as NCT00331032, 29 May 2006) was carried out in the USA and Kenya as a randomized, placebo-controlled trial. The conclusion was that genitourinary adverse events and colposcopy findings were consistent with mild epithelial irritation and inflammation, in particular when using a 3% dosage [28]. It was also shown in this study that several mucosal immune parameters were increased compared to the placebo [29]. Another phase 1 clinical trial (registered as NCT00442910, 5 March 2007) was carried out almost at the same time in the USA (San Juan, Puerto Rico and Tampa, Florida). The conclusion was that VivaGel^®^ was generally well-tolerated and comparable with the placebo although there was a higher incidence of low-grade related genital adverse effects, as in the previous study [30]. Another conclusion concerned the need for improvement of gel formulations to seek better acceptability [31]. The tolerance by men [32] and its safety for them (Phase 1 NCT00370357, 31 August 2006) were also assessed [33].

The antiviral effects were early assessed, in particular against genital herpes (Phase 1 NCT00331032, 29 May 2006) and against HIV (Phase 1 NCT00370357, 31 August 2006, and Phase 2 NCT00740584, 25 August 2008) [34]. However, the clinical trials were then oriented towards the study of the anti-bacterial properties of SPL-7013 [35], in particular against bacterial vaginosis (BV). A phase 2 clinical trial (NCT01201057, 14 September 2010) was conducted in a double-blind, multicenter, randomized, placebo-controlled, dose-ranging study for the treatment of bacterial vaginosis. It was shown that VivaGel (named Astodrimer in this study and in the followings) once daily for 7 days was superior to placebo and was well-tolerated at 1% dose, supporting a role for such gel as an effective treatment for bacterial vaginosis [36]. The dose range and the efficiency of the treatment against bacterial vaginosis was then assessed in several phase 3 clinical trials. Several publications have presented in details the outcomes of these clinical trials [37]. Two phase 3 clinical trials were carried out in the USA for one (NCT01577238, 13 April 2012) and in the USA, Germany, and Belgium for the second one (NCT01577537, 16 April 2012, and 2012-000752-33, 22 June 2012). Both studies confirmed that Astodrimer is an effective, safe, and well-tolerated treatment for women with bacterial vaginosis [38]. Another phase 3 study (NCT02237950, 12 September 2014, and 2014-000694-39, 21 October 2014) demonstrated that the gel administered every second day for 16 weeks was effective and superior to placebo for prevention of recurrent bacterial vaginosis in women with a history of recurrent BV [39]. A recent review emphasized that such formulation inhibits growth of bacteria associated with BV via a novel mechanism of action, compared with conventional antibiotics, by blocking the attachment of bacteria to cells and inhibiting the formation of and disrupting biofilms [40]. Table 1 displays the different types of clinical trials carried out with VivaGel^®^.

VivaGel^®^ is now available in different countries under different names, as gel against BV, and also as lubricant for condoms, as it has been shown in laboratory studies to inactivate HIV, herpes simplex virus (HSV), and human papillomavirus (HPV) [26]. A recent and important sequel of the antivirus properties of dendrimer SPL-7013 concerns its efficiency against COVID-19. It has been reformulated in the form of a nasal spray to prevent COVID-19 infections under the name VIRALEZE^TM^ [41]. Tests were carried out on healthy volunteers at an Australian clinical trials facility [42].

Another example of poly-L-lysine dendrimer in the port-folio of Starpharma is numbered AZD0466. It is a fifth-generation poly-L-lysine dendrimer bearing two types of terminal functions, 32 PEG_2100_ molecules and 32 molecules of a dual Bcl-2/Bcl-x_L_ inhibitor (anti-tumor agent) named AZD4320, attached through the linker X = N-Me (this compound is also named SPL-8977). Other different linkers have been used to modify the release rate of the drug (X = CH_2_ for SPL-8931, X = S for SPL-8932, and X = O for SPL-8933) (Figure 3). The slowest release was observed with SPL-8931, which displayed no cardiac toxicity, contrarily to the compound having a rapid release. Combination of modeling and experiments finally converged to the choice of AZD0466 (SPL-8977, X = N-Me) as the best choice [43]. This dendrimer was tested on mouse xenografts of malignant pleural mesothelioma. AZD0466 was as effective as the standard-of-care chemotherapy, Cisplatin, and displayed significantly reduced degree of thrombocytopenia [44]. AZD0466 was also shown suitable to be delivered subcutaneously; it can act as a circulating drug depot accessing both the lymphatic and blood circulatory systems in mice to fight against lymphoma [45].

In view of these interesting pre-clinical results in mice, at least three clinical trials are presently underway with AZD0466. NCT04214093 (30 December 2019) is a phase 1, first in human study in patients with advanced solid tumors, lymphoma, multiple myeloma, or hematologic malignancies. NCT04865419 (29 April 2021) is a phase 1 (AZD0466 alone) and phase 2 (AZD0466 in combination with the drug voriconazole) study in patients with hematological malignancies. NCT05205161 (25 January 2022) is a phase 1/phase 2 in patients with non-Hodgkin lymphoma.

The same concept of dually functionalized polylysine dendrimers was also applied to the docetaxel-dendrimer conjugate SPL8783, tested in the phase 1 clinical trial EudraCT Number 2016-000877-19 (13 July 2016) in patients with advanced solid tumors [46]. Another example in the same field concerns cabazitaxel–dendrimer conjugate CTX-SPL9111, named DEP®-SN38 [46]. The phase 1/2 clinical trial EudraCT Number 2017-003424-76 also concerned patients with advanced solid tumors. The phase 1/2 clinical trial EudraCT Number 2019-001318-40 was carried out with the same aims [23].

In relation with polylysine dendrimers, a less defined “Dendrigraft” fifth-generation polylysine [47] was mixed with nitro-imidazole-methyl-1,2,3-triazol-methyl-di-(2-pycolyl) amine. This mixture was named ImDendrim and was found suitable for the encapsulation of rhenium and radioactive ^188^Re (Figure 4). This potential radiopharmaceutical agent displayed anti-tumoral activity in liver cancer in mice [48]. The clinical trial NCT03255343 (21 August 2017) with [^188^Re]rhenium-ImDendrim concerns patients with non-operable liver tumor with resistance to other classical chemotherapy.

## 3. Clinical Trials with PAMAM Dendrimers

PAMAM (PolyAMidoAMine) dendrimers [16] are the most widely used type of dendrimers. In the native form, they have NH_2_ terminal functions as poly-L-lysine dendrimers; thus, they can be easily modified at will. However, the risk of reverse Michael addition, particularly in acidic conditions, inducing the breakage of the structure, has precluded for a long time the use of PAMAM dendrimers in clinical trials [37]. Nevertheless, PAMAM dendrimers have been recently tested in clinical trial, essentially proposed by Ashvattha Therapeutics [49]. All were obtained by first modifying the terminal functions to hydroxyl, then by stochastically grafting other functions. The first dendrimer, labelled OP-101, is a fourth-generation PAMAM stochastically functionalized with 40 OH and 24 N-acetyl cysteine functions. Figure 5 displays the full structure of one of the possible isomers; Figure 6A displays a very simplified structure of the same compound.

It was shown that this compound OP-101 has anti-inflammatory and anti-oxidant activity in microglia cells [50]. The first clinical trial, NCT03500627 (18 April 2018), was a phase 1 study to evaluate the safety, tolerability, and pharmacokinetics of OP-101 injected intravenously. The second trial, NCT04321980 (26 March 2020), evaluated the subcutaneous administration of OP-101. The objectives of the phase 2 study NCT04458298 (7 July 2020) are to evaluate the efficacy of OP-101 in patients with severe COVID-19 and its effect in reducing pro-inflammatory cytokines biomarkers. No drug-related adverse events were reported. OP-101 reduced morbidity and mortality in hospitalized patients with severe COVID-19. Indeed, at day 60, only three of seven patients given placebo were surviving, whereas they were 14 of 17 for OP-101–treated patients [51].

A second example of PAMAM hydroxyl dendrimers stochastically functionalized concerned the grafting of known drugs suitable for treating diverse eye diseases. A recent paper published this concept using triamcinolone-acetonide as drug conjugated to the dendrimer and tested in models of age-related macular degeneration [52]. However, the drug presently under clinical trials is a generation-four PAMAM hydroxyl dendrimer partly functionalized with Sunitinib, an inhibitor of vascular endothelial growth factor receptors (simplified structure in Figure 6B). The grafting through an ester bond enables the slow release of the drug. This dendrimer, named D-4517-2 [53], has been evaluated in the phase 1 clinical trial NCT05105607 (3 November 2021) for safety and tolerability after subcutaneous injection. The phase 2 NCT05387837 (24 May 2022) intends to evaluate safety, tolerability, and pharmacokinetics in patients with neovascular (wet) age-related macular degeneration or subjects with diabetic macular edema.

A third example of PAMAM-hydroxyl dendrimer concerns the stochastic grafting of a few radionuclides, in particular ^18^F, for specific positron emission tomography (PET) imaging and radiotherapy, with the aim of detecting inflammatory sites or tumors [54]. This specific dendrimer is named 18F-OP-801 (simplified structure in Figure 6C) and is used in the phase 1 clinical trial NCT05395624 (27 May 2022) to evaluate the safety, pharmacokinetics, and biodistribution after intravenous administration to patients with amyotrophic lateral sclerosis.

Another clinical trial with PAMAM dendrimers is NCT04262076 (10 February 2020), which concerns the use of a PAMAM dendrimer with Pulpine for the remineralization of carious-affected dentin. However, the generation and the type of surface functions of the PAMAM dendrimer used are unknown in this study carried out by the Al-Azhar University (Egypt). No result has been posted.

## 4. Clinical Trials with Other Types of Dendrimers

Two clinical trials against COVID-19 are currently on-going, both based on the same cationic peptide dendritic structure having lysine as branching elements, named KK-46, whose structure is shown in Figure 7 [55]. This compound is used as carrier of siRNA modified for silencing SARS-CoV-2 (siCoV) by inhibiting its replication [56]. The association of KK-46 with siCoV is named MIR 19^®^. Trial number NCT05208996 (26 January 2022) is a phase 1 clinical trial to assess the safety and tolerability of siCoV/KK46 in healthy volunteers. Trial number NCT05184127 (11 January 2022) is a phase 2 clinical trial to assess the efficacy and safety of MIR 19^®^ via 14 days of treatment of hospitalized patients with symptomatic moderate COVID-19.

## 5. Discussion

This review is the first one fully dedicated to clinical trials carried out with dendrimers. It can be seen by the dates on which they have been posted that many of them are extremely recent (dated 2020–2022); thus, most of them concern phase 1 assays. The only dendrimer under phase 3 clinical trials is the polylysine used as an active ingredient of VivaGel^®^ (Figure 2B) to treat bacterial vaginosis. It must be emphasized that the phases 1 with this dendrimer were carried out in 2004, 2006, and 2007, whereas the phases 3 began in 2012 and 2014, almost 10 years after the beginning of phases 1. Furthermore, VivaGel^®^ is a gel, which does not penetrate the skin or the bloodstream, and thus cannot cause serious health problems. On the contrary, most of the dendrimers recently used in phase 1 clinical trials were injected either intravenously or subcutaneously. As practically no results have been posted with most of them, it is difficult to foresee if and when phase 3 clinical trials could be carried out with them. It is even more difficult to foresee if they could be used in personalized medicine.

However, dendrimers have already found application in personalized medicine to enhance the efficiency of diagnosis tools [57]. They are used as 3D spacers in chips for bioassays between the solid surface and the probe to increase the binding efficiency of the probe. Indeed, an increased distance from the solid surface enables the trapping of the target as efficiently as in solution, as illustrated in Figure 8. Such a concept was applied very early with PAMAM dendrimers for enhancing the radial partition in immunoassays. It is used in particular in the Stratus® CS Analyzer * for acute care diagnostics, which provides quantitative cardiac assays for fast, accurate evaluation of patients presenting suspected myocardial ischemia [58]. Another type of bioassays is carried out with phosphorhydrazone dendrimers [59]. Different types of chips are elaborated depending on the disease to be analyzed [60]. DendrisCHIP^®^RD concerns the simultaneous detection of 10 bacteria commonly involved in respiratory infections. DendrisCHIP^®^OA allows the detection of 10 bacteria responsible for deep and osteoarticular infections [61]. DendrisCHIP^®^DP is dedicated to the identification of fungi involved in dermatophytosis, whereas DendrisCHIP^®^SD is dedicated to the identification of 10 pathogens (bacteria, yeasts, and viruses) responsible for sexually transmitted infections [60].

## 6. Conclusions and Perspectives

The huge number of dendrimers already synthesized and the even larger number of dendrimers that could be synthesized as active *per se* or as carriers of drugs should have generated breakthrough in medicine, including in personalized medicine. However, this is not the case, with two exceptions that are bioassays for the specific detection of diseases and gels for the local treatment of bacterial vaginosis. The hope of expanding the number of dendrimers actually used in human health lies in the phase 1 clinical trials carried out recently and on others that are planned. One can cite in particular ongoing and future assays against influenza A and B virus; against COVID-19 [26]; against neurology, ophthalmology, inflammatory diseases, and neuro-oncology [49]; and to modulate the activity of the immune system to treat chronic inflammatory diseases [62,63]. The field of theranostic (diagnosis and therapy with the same “object”) could be developed with multifunctional dendrimers (see for instance 18F-OP-801). Thus, there are still many pending works, but I do believe that dendrimers can be considered as an emerging opportunity in medicine and in part in personalized medicine.

## Figures and Tables

**Figure 1 jpm-12-01334-f001:**
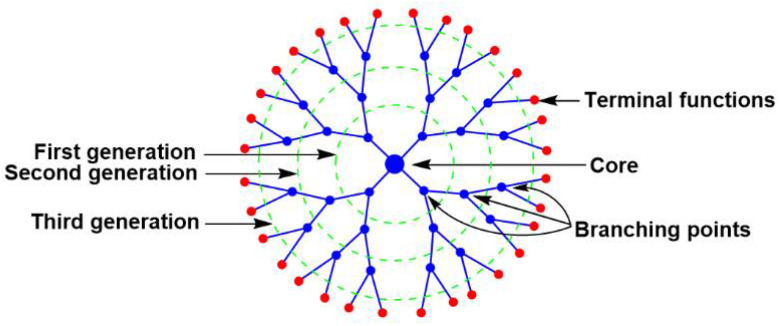
Schematized third-generation dendrimer, showing its main structural features.

**Figure 2 jpm-12-01334-f002:**
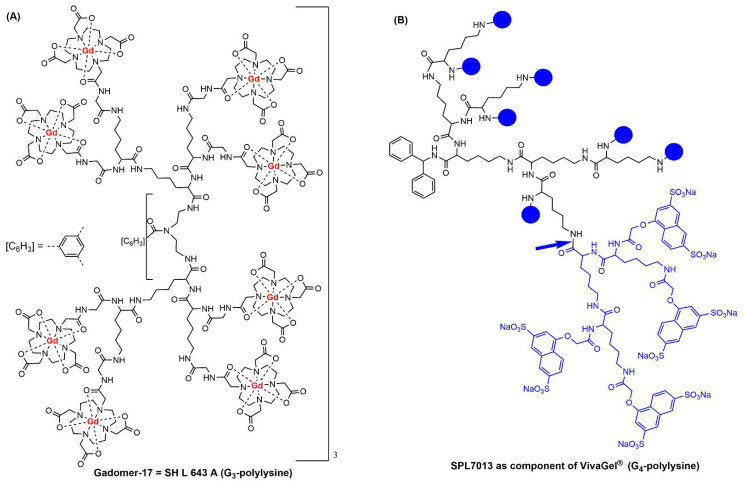
Chemical structure of two poly-L-Lysine dendrimers. (**A**) Gadomer-17 used as MRI agent. (**B**) SPL7013, also named Astodrimer sodium, the active ingredient in the mucoadhesive gel VivaGel^®^; the blue bowls correspond to the structure that is shown in blue after the blue arrow.

**Figure 3 jpm-12-01334-f003:**
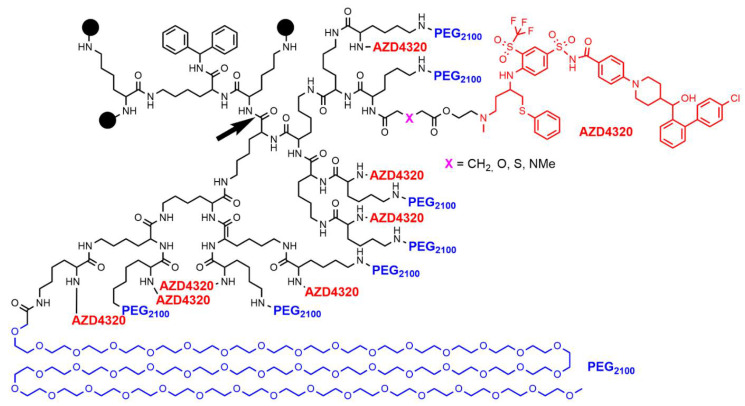
Structure of dendrimer AZD0466 (X = N-Me). The black bowls represent the structure that is shown after the black arrow.

**Figure 4 jpm-12-01334-f004:**
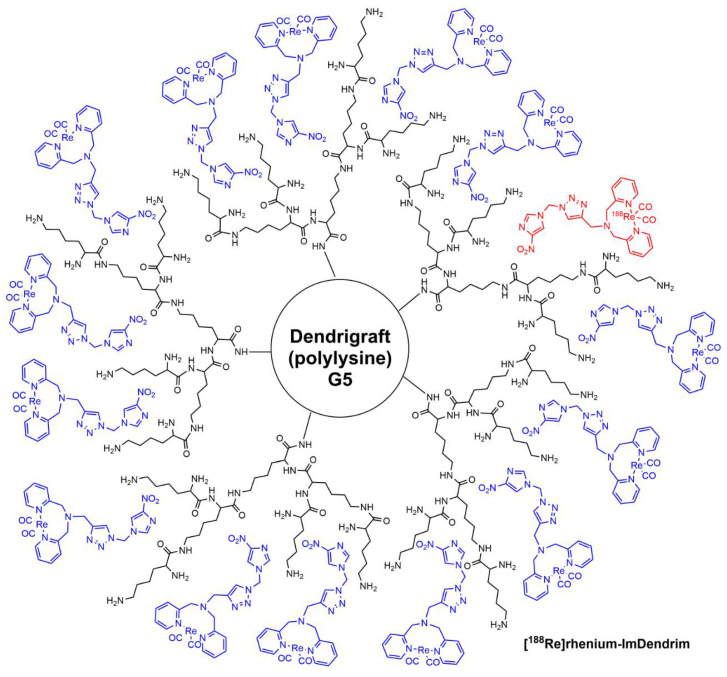
Schematized structure of [^188^Re]rhenium-ImDendrim.

**Figure 5 jpm-12-01334-f005:**
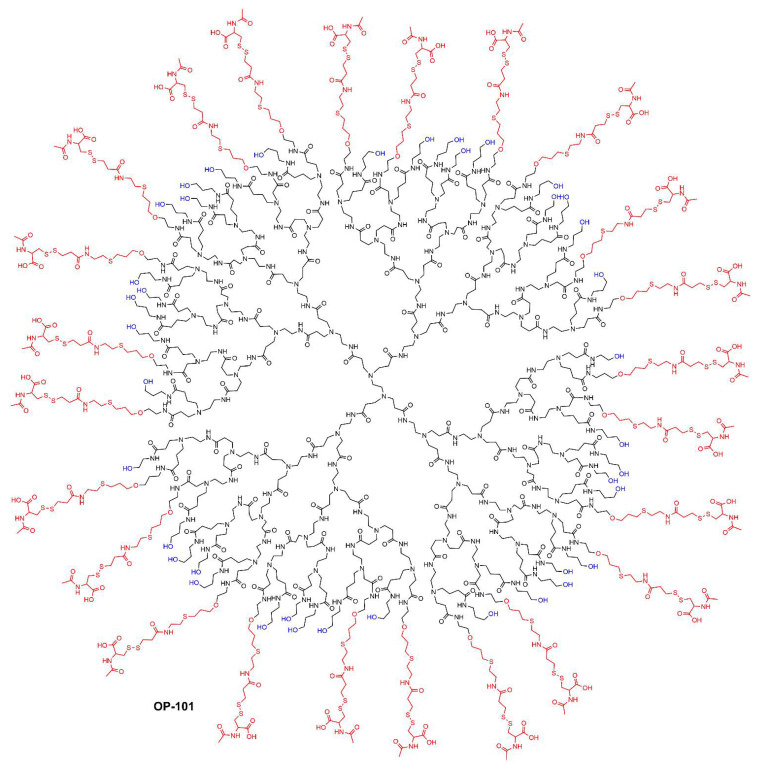
Hydroxyl generation 4 PAMAM dendrimers stochastically functionalized; one of the possible full structure of OP-101.

**Figure 6 jpm-12-01334-f006:**
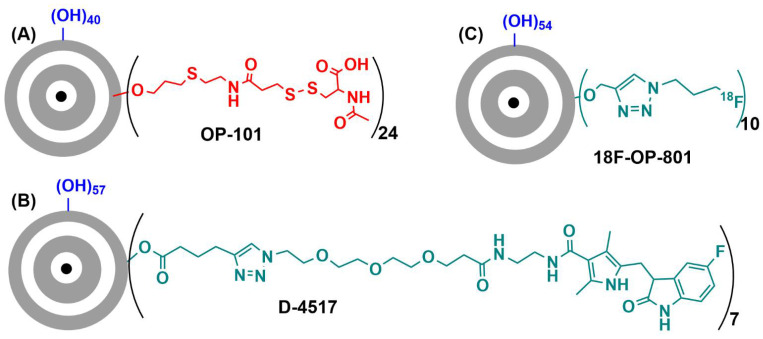
(**A**) Simplified structure of OP-101. (**B**) Simplified structure of D-4517. (**C**) Simplified structure of 18F-OP-801.

**Figure 7 jpm-12-01334-f007:**
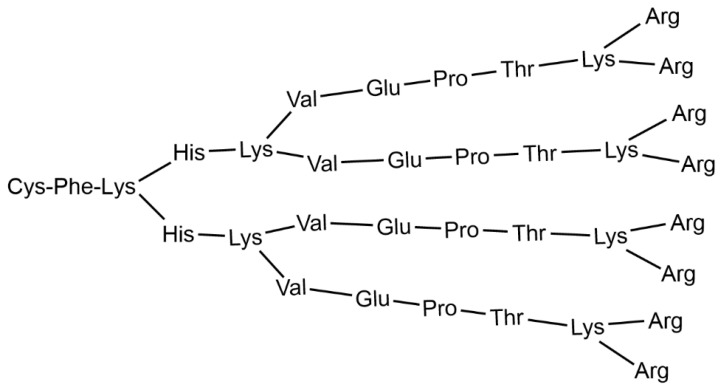
Structure of the cationic peptide dendrimer KK-46, usable as carrier of siCoV.

**Figure 8 jpm-12-01334-f008:**
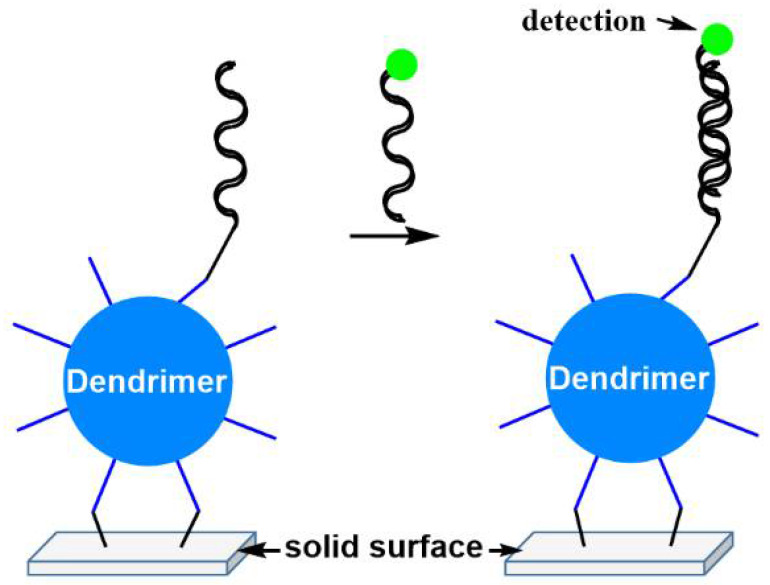
Principle of the use of dendrimers in bioassays.

**Table 1 jpm-12-01334-t001:** Clinical trials with VivaGel®. Information taken from https://ClinicalTrials.gov (accessed on 2 August 2022) for NCT numbers and from https://www.clinicaltrialsregister.eu (accessed on 2 August 2022) for the other numbers.

Study Number	Date Posted	Phase	Aims
NCT00331032	29 May 2006	1	Safety and tolerability (female)
NCT00370357	31 August 2006	1	Safety (male)
NCT00442910	5 March 2007	1	Safety and acceptability (female)
NCT00490152	22 June 2007	1	Adherence, acceptability (female)
NCT00740584	25 August 2008	1/2	Retention and duration of activity (female)
NCT01201057	14 September 2010	2	Efficacy against BV ^1^ (female)
NCT01437722	21 September 2011	2	Prevention of recurrence of BV ^1^ (female)
NCT01577238	13 April 2012	3	Treatment of BV ^1^ (female)
NCT01577537	16 April 2012	3	Treatment of BV ^1^ (female)
2012-000752-33	22 June 2012	3	Treatment of BV ^1^ (female)
NCT02236156	10 September 2014	3	Prevention of recurrence of BV ^1^ (female)
NCT02237950	12 September 2014	3	Prevention of recurrence of BV ^1^ (female)
2014-000694-39	21 October 2014	3	Prevention of recurrence of BV ^1^ (female)

^1^ Bacterial vaginosis.

## Data Availability

Not applicable.

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
