# Peer review of "Dendrimers, an Emerging Opportunity in Personalized Medicine?"

_jpm, 2022, doi:10.3390/jpm12081334_

Round 1

Reviewer 1 Report

In the present manuscript, A.M. Caminade focusses on the clinical trials carried out with three types (polylysine, polyamidoamide (PAMAM) and polypeptide) dendrimers. After informative introduction, author starts with formulations based on polylysine dendrimers and above all VivaGel as successful product followed by AZD0466 as very promising candidate. PAMAM dendrimers are topic of next capture and to finish the interest is focused on polycationic peptide dendrimers aimed against Covid-19.

This is a well written and very informative paper. I think, this manuscript could be interesting for scientist in the field. I only have a one point to resolve:

Line  120-122:  In the text, there is a little bit confusing sentence “….dual Bcl-2/Bcl-xL inhibitor (anti-tumor agent) named AZD4320, attached through different linkers X (X = CH2 for SPL-8931, X = S for SPL-8932, X = O for SPL-8933 and X = N-Me for SPL-8977)….“, but only SPL-8977 derivative is (acording to ref. 43 in the manuscript) denoted as AZD0466 and not all analogue as can result from text.

Author Response

I thank this reviewer for his/her useful comments.

I agree that this sentence is rather confusing, even if it is true. It is due to the fact that the structure of AZD0466 is rather confusing in the paper (ref. 43). Anyway, I have modified this sentence, and another one, to be coherent.

Reviewer 2 Report

This is a much needed review about clinical trials of drugs or carriers based on dendrimers. There are oceans of literature describing all sorts of potential uses of dendrimers, but in the end, the translation into the clinic is the problem.

Author Response

I thank this reviewer for his/her very positive comments.

This reviewer has no query.